# Beyond tingles: An exploratory qualitative study of the Autonomous Sensory Meridian Response (ASMR)

**Enya Autumn Trenholm-Jensen**[1], **Lindsey Burns**[2], **Jill Ellen Trenholm**[3], **Christopher James Hand**[4]*

**1** Department of Psychology, Glasgow Caledonian University, Glasgow, United Kingdom, **2** Department of Psychology, Heriot-Watt University, Edinburgh, United Kingdom, **3** Department of Women's and Children's Health, Uppsala University, Uppsala, Sweden, **4** School of Education, University of Glasgow, Glasgow, United Kingdom

* Christopher.Hand@glasgow.ac.uk

**Data Availability Statement:** Raw data are audio-video recordings of participant interviews. Thus, it is possible to identify participants from these recordings and it is not appropriate to share such

## Abstract

This qualitative exploratory study investigated the embodied experiences and the meanings of Autonomous Sensory Meridian Response (ASMR) from the viewer's perspective. ASMR research has been sparse and largely quantitative, assuming it to be a predominantly fixed physiological response of "tingles", despite the acronym being rooted in pseudoscience. A qualitative research design was adopted to facilitate the exploratory nature of the study in this under-researched area. In contrast to the mostly survey-based research on ASMR, this study employed semi-structured interviews as a means to understand the lived experience of ASMR and to promote participant agency. Six self-identifying ASMR consumers were recruited using a mixture of snowball and opportunity sampling. Semi-structured interviews were conducted both in person and using Skype to facilitate transnational data collection. Interview transcripts were analysed using an inductive, data-driven approach to thematic analysis. The analysis suggests that ASMR is felt to provide a social environment of comfort rather than a solely physiological-based experience. Three key themes emerged: *who and what defines ASMR*? (reflecting the variety of what was classed as ASMR and what content was consumed to produce the response); "*real" intimacy tailored to me* (reflecting the idiosyncratic perception of intimacy made possible through ASMR)*;* and *emotional relief on my terms* (reflecting the role of ASMR in self-soothing). The present data reflect a rich, complex experience of the ASMR consumer, pointing to potential wider applications and informing further research.

## Introduction

Even prior to the COVID-19 pandemic face-to-face contact was becoming less prevalent in daily interpersonal exchanges, particularly in a Western context [1]. At the same time, avenues for socialising are being enabled through technology, particularly through the development and ubiquitous use of social media and chat apps [2]. The inherent social nature of humans continues to be expressed in new ways via the internet, one of which is proposed to be through

raw data. We have provided access to transcribed data: https://osf.io/9ur3m/.

**Funding:** The author(s) received no specific funding for this work.

**Competing interests:** The authors have declared that no competing interests exist.

*autonomous sensory meridian response* (ASMR) content creation and consumption. These new ways of socialising may well remain long after the pandemic necessitates them, suggesting a need for research to better understand them.

ASMR is the term used to describe a physiological reaction to a range of audio-visual and tactile triggers. ASMR is anecdotally reported to comprise a tingling sensation around the skull which travels down the spine to promote a sense of wellbeing and intense relaxation [3]. Prior to being known as ASMR, it was referred to as a 'brain orgasm' [4] as well as an 'attention-induced head orgasm' underscoring the sensation felt [5]. The first recorded discussion of the sensation that is now referred to as ASMR was on an internet health forum in 2007 [6]. Later in 2010, Jennifer Allen, a member of the burgeoning online community, coined the term ASMR and started a Facebook page dedicated to it [7]. As the focus of this paper is to understand the user experience, the term 'ASMR consumers' was selected as the most adequate term. This is not to be confused with those who experience ASMR but do not consume it or vice versa. The purpose of this study is to hear from individual members of the ASMR community itself and attempt to understand what drives their conscious consumption of ASMR.

The ASMR acronym consists of *Autonomous*, which refers to the reflexive nature of the reaction while *sensory* alludes to the physiological factors that enable it. *Meridian response* is a euphemism for orgasm [7]. In an interview with the leading website on ASMR, ASMR University, Allen cited the importance of developing a term for the response which sounded technical and clinical to discourage embarrassment among the community and to encourage sharing of experiences. Allen also maintained that the term serves a purpose in giving the response scientific appeal and in that way encourages empirical research into this topic [7]. The etymology of the acronym demonstrates how ASMR has been as much about the community, and enabling its creation and maintenance, as it has been about the sensation since it begun. The first peer-reviewed research paper on ASMR was published in 2015 [3].

ASMR has blossomed into an international phenomenon. On average, eleven videos with the hashtag #ASMR were uploaded to YouTube per hour in 2015 [8]. The increased demand can also be felt through the amount of YouTube searches for "ASMR", which have roughly tripled between 2016 and 2020 [9]. Currently, the three largest channels dedicated to ASMR content on YouTube boast a combined seven million subscribers and average 36 million views each month [10]. Videos dedicated to eliciting the response are broadly categorised into two types: *intentional* and *unintentional*. Intentional ASMR videos predominantly take the format of the seated "ASMRtist" [11] looking into the camera and performing various auditory-focussed acts such as whispering, tapping, and chewing, whereas unintentional ASMR as the term suggests refers to videos or other auditory stimuli which prompts the response inadvertently. Some of the videos in this category include Bob Ross painting tutorials and natural sounds such as rain [12]. Our initial preliminary literature search returned fewer than 35 published academic articles in this area. Of these articles, none were solely qualitative.

Attempting to clarify the various ways in which ASMR can be understood, Hostler et al. [13] distinguished between state and trait ASMR. The former refers to the emotional experience of ASMR and the latter to the reliable triggering of the physiological experience. However helpful, this distinction is difficult to establish outside of a lab and is also susceptible to considerable overlap. Nevertheless, the current study pursues a better understanding of state ASMR.

Due to the pseudo-scientific roots of ASMR, the bulk of research in this area aimed to establish the physiological validity of the response. Poerio et al. [14] found consistent reductions in heart rate during the viewing and listening of an ASMR video by 112 self-identified ASMR experiencers. The same study also found increased skin conductance levels for the ASMR group. The lowered heart rate, coupled with increased skin conductance, revealed a simultaneous activating and deactivating response, appearing intuitively erroneous. However, similar

effects have been noted in research of for example, nostalgia–whereby a neural basis for the mixture of joyful and sad affect was conjured [15, 16].

Experiencers of ASMR have also been shown to have high ratings of neuroticism [17]. Individuals who score high on neuroticism are more likely to experience anxiety, a trait which has been found to be associated with increased heart rate variability (HRV) [18]. HRV can depend on factors such as anxiety and perceived stress [19–21]. This variability is not detected in baseline measurements as HRV does not entail a higher or lower heart rate generally, but rather involves more dramatic fluctuations in heart rate in response to stimulus [22]. Fluctuations in HR may depend on trait anxiety in participants who are drawn to ASMR rather than as a direct result of consuming ASMR. This notion is consistent with current research that posits ASMR as an emotional regulation tool rather than a sensory response trigger [23].

Lochte et al. [24] took a neuroscientific approach to observe the neural correlates underlying ASMR. Using fMRI, they observed a significant activation of the medial prefrontal cortex (mPFC) during the moments of tingling while watching an ASMR video [24]. The mPFC is part of a network associated with social behaviours and cognitions. From these findings it was extrapolated that ASMR may engage the brain in a similar way to socialising [25–27]. Undergoing the audio-visual experience of ASMR is predominantly a solitary activity as self-reported by users [3]. However, ASMR as a replication of a real-world interpersonal socialisation contradicts the solitary element of the experience. The mPFC typically represents the former, however, ASMR is predominantly computer-mediated and socially minimal bearing similarity to a parasocial interaction (the AMSR users' reaction to and perceived relationship with the ASMRtist) [28] more than a real-life one. The parasocial intimacy reported and experienced is likely exacerbated by the one-on-one experience with the ASMRtist. This is further enabled by the ASMRtist who themselves also feature alone in their videos. Therefore, ASMR may be an intimate encounter rather than a solitary activity [29]. The activation of the mPFC may also suggest a release of oxytocin which has been linked to reduced anxiety [30, 31]. Furthermore, the combination of a social presence and the release of oxytocin has been found to dampen fear producing a calming effect [32]. This hormone has also been found to play an enhancing role in the connectivity between the mPFC and the amygdala [33, 34]. As these two regions interact during social behaviours and emotion regulation, this could be a further example of the neural similarity between experiencing ASMR and socialising, and the possible neuro-chemical benefits resultant of both/either. The release of oxytocin due to ASMR triggers is, however, still speculative.

An fMRI investigation of the ASMR reaction by Smith et al. [35] supported the findings of Lochte et al. [24], with significant activity in the prefrontal regions of the brain. In addition, significant activity was observed in the primary motor cortex, the auditory cortex and the occipital lobe which is the centre for visual processing in the brain [35]. This range of activation suggests that ASMR is both emotional and sensorimotor in nature. An added noteworthy discovery of both studies was the fact ASMR was able to be triggered in the noisy fMRI setting. Previously, it was reported to be an activity which required silence and solitude to elicit the stimulation [3].

Physiological measures for ASMR have also been studied in the form of pupil dilation. Valtakari et al [36] employed eye tracking to examine whether ASMR would be detectable through measurements of pupil diameter. Changes in pupil dilation were only detectable during periods of tingling as self-reported by the ASMR participants, not throughout the ASMR video stimulus in general. These results support the physiological basis of the tingles and are also in line with the study which identified a rise in skin conductance as both responses are linked to physiological arousal [14, 37]. Valtakari et al. [36] sampled ASMR consumers as well as a non-user control group. Throughout the course of the study, members of each group reported

feeling unsure if they were experiencing ASMR. Tingles are one of the main features of ASMR which are often treated as a metric of stimulation in this field of research. It should be noted, however, that tingles are a poorly defined criteria and can lead to the type of ambiguity of experience that was found by Valtakari et al. [36]. This is of additional importance to the current study which aims to understand the experience of those who consume ASMR. The current physiological definition accounts for the response in its purest form but does not cover the interaction of ASMR consumers and AMSR content. Much is left unknown in terms of whether the response is in fact the main motivator for their consumption.

Additional attempts to understand ASMR have explored personality traits associated with ASMR consumers. Fredborg et al. [17] used a Big Five Inventory (BFI) [38] as well as an ASMR checklist to explore these associations. Results showed higher ratings of Openness to Experience for ASMR participants versus the control group, and higher yet for female ASMR participants. Fredborg et al. [17] also found greater Introversion (lower Extraversion) in the ASMR group versus controls. Introversion is often associated with a lower threshold for arousal in social settings [39, 40]. There is a plausible connection between the aforementioned neural similarity of ASMR and socialising with introverts' lower threshold for social stimulation. Roberts et al. [41], however, suggests that extraversion and neuroticism may be associated with the intensity of the ASMR experience rather than the general propensity. Using the Highly Sensitive Person Scale in conjunction with the BFI, Roberts et al. [41] also argued that higher overall dispositional sensitivity could account for the heightened sensory arousal in ASMR experiencers rather than particular personality traits, where only weak correlations were found. Janik McErlean and Banissy [42] examined ASMR and empathy using the Inter-Personal-Reactivity index (IRI) [43] and the BFI [38]. Their results revealed that ASMR consumers scored higher on the Empathic Concern and the Fantasising subscale of the IRI. According to the implications of the scale, this signifies ASMR consumers as being inclined to have increased levels of sympathy and compassion. The Fantasising element suggests an ability to identify with actions and emotions of fictional characters, thus further supporting the aforementioned parasocial element. Support for this claim was found in a study by Keizer et al. [44] who found those who experience ASMR are more prone to illusory sensory events than controls. Keizer et al. [44] suggest that ASMR participants are delineated in not only their imagery ability but also their physical experience of events as measured by the Sensory Suggestibility Scale [45]. Further support for these findings was found by Smith et al. [46] who measured the functional connectivity in the default mode network of participants self-reported to experience ASMR and controls. They identified a reduced ability to inhibit sensory emotional experiences for ASMR experiencers that would commonly be suppressed in others. Moreover, a similar fantasising ability is required to enjoy roleplay [47, 48]. It is no coincidence that roleplay is one of the most common themes in ASMR videos, ranging from health-care professionals to fictional characters. Similar to Fredborg et al. [17], participants rated higher on the measure of Openness to Experience. These factors have conceptual similarities in terms of how Openness to Experience can beget the ability to fantasise [49, 50].

Other attempts to understand ASMR have been made by comparing it to several other sound-based conditions and reactions. A recent study [51] related ASMR to frisson, a term often used by musicologists to explain an intensely pleasurable and visceral reaction to music [52]. Both frisson and ASMR have been documented to prompt pupil dilation in individuals demonstrating a shared physiological manifestation of both sensory experiences [36]. However, an increased heart rate was elicited in individuals experiencing frisson, whereas ASMR consumers were found to have a lowered heart rate when consuming ASMR [14]. Moreover, ASMR is mainly reported as a tingling sensation whereas frisson has been reported to provoke chills. Interestingly, both sensations are usually localised to the head, spine and torso, often

initiating at the former and spreading to the latter. Despite this similarity, studies in lab settings have found that participants were able to distinguish between frisson and ASMR, suggesting they provide noticeably distinct physiological experiences [17, 41]. An additional link between the concepts is demonstrated through the personality traits associated with both phenomena. Two separate studies, one on ASMR [17] and one on frisson [53], both identified a higher rating of Openness to Experience as measured by the BFI, suggesting a multifaceted connection between the distinctive experiencing of certain sounds and personality features. Interestingly, these studies relied on self-report regarding experiencing frisson or ASMR. Furthermore, binaural sound, the type of audio which features in the majority of ASMR recordings, has also been found to induce subjective frisson [54] indicating an additional similarity between the two audio-induced sensory concepts. ASMR has also been linked to misophonia. In contrast to frisson, misophonia denotes the negative emotional, behavioural and physiological responses to certain sounds [55]. Using a misophonia questionnaire [56] with self-identifying ASMR users, Janik McErlean and Banissy [57] identified heightened rates of misophonia amongst them. Therefore, it was theorised that ASMR and misophonia may exist on the same spectrum of sound sensitivity but reside on opposite ends.

Although sound features prominently in the production and experience of ASMR, the importance of this aspect of the phenomenon is not clear. In a study by Janik McErlean and Banissy [42], the sampling process recognised participants as experiencers of ASMR through aligning with an audio-focused definition of ASMR. Despite identifying with the definition, the results indicated that the third most-important factor for stimulating ASMR was not a sound but the *act* of personal attention. Personal attention was also ranked as the fourth most important factor in ASMR in a separate study by Poerio et al. [14]. Despite attempting to narrow down the definition of ASMR to a sound-based experience, the study still gave rise to results indicating ASMR to be a multi-sensory experience. The types of videos that were reported to induce the personal attention factor were mainly roleplay videos such as doctor/patient and office roleplays.

An alternative view of sound in the context of ASMR is how sound could be a vehicle for interpersonal connection. Waldron [58] argued that as opposed to sound being the catalyst for the ASMR experience, it merely plays a suggestive role in indicating intimacy between producer and consumer. Similarly, the use of sound technology such as binaural microphones and the distinct lack of background noise in ASMR creates a soundscape which indicates an environment of intimacy. Beyond whispering and speaking, Gallagher [59] argues that there is intimacy in the acts of crinkling paper and tapping inanimate objects. These sounds recorded in high definition are referred to as *materialising sound indices* in film-making and are colloquially referred to as the sounds that make us *feel* [60]. By this, it is meant that these sounds put us in touch with the materiality of the sound source thereby causing us to feel an association with the source [60]. This point is furthered by Smith and Snider [61] who argue that ASMR underscores the ability for mundane sounds to create affective experiences.

Andersen [62] argues that the social factors involved in ASMR are no less important than sound for stimulation of this response. The ASMR acronym does not include any terms which allude to a social interaction, however most videos with the ASMR label involve some form of social currency, for example eye contact and proximity [61]. Andersen [62] illustrated the distant intimacy that is fundamental to most ASMR videos by outlining the role of expectation on behalf of the ASMR experiencer. As most ASMR consumers report using it to aid relaxation or facilitate falling asleep, Andersen [62] contends that this predisposition enables the sensory experience through intention. This could be related to expectation priming and a pseudo placebo effect, where expectation of relaxation can trigger the desired response regardless of the stimuli due to priming [63] thus, resulting in a placebo-type effect [64]. This connection to

ASMR is tentative, though the validity of the priming and placebo-type effects have been empirically validated [65–68]. Further studies have found that the effect of expectation priming has enhanced placebo-type effects [69, 70]. As the ASMR community is largely dedicated to conscious creation and stimulation of the response, the role of intention and the argument for an expectancy effect may have merit. The same cannot be said for unintentional ASMR; many experiencers of ASMR report having their first encounter of ASMR before they know the meaning of the term [7]. In this way it is difficult to argue the predisposition for the response.

A case study on an ASMR radio show contended that the appeal of ASMR stemmed from what the author calls the 'reassuring female voice' [71]. This voice symbolises the presence of a maternal figure or the care-taking role often assumed by women [72, 73]. With most ASMR practitioners being female, this is a reasonable assumption in explaining the draw of these videos, though it does not necessarily apply to genderless unintentional ASMR such as rain sounds. Bjelić [72] furthers the argument for the socially intimate nature of ASMR, where the digital intimacy of ASMR was addressed. They contended that ASMR videos are meeting a timeless need in a novel way. Bjelić [72] suggested that in our current digitised environment, more people are turning to the internet for advice and care and ASMR fulfils a desire in the viewer to be attended to, cared for, and valued. This could be especially relevant given the current context of the pandemic, and this interpretation of ASMR as being therapeutic and providing relief from stresses in daily life has been supported by for example O'Connell [74] and Maslen and Roache [75]. O'Connell [74] contends that the content is purposefully mundane and uneventful because the main attraction is not the activity performed but the attention that is being paid to the viewer.

There is a vibrant ASMR community of viewers as evidenced by the millions of followers that researchers know very little about. There seems to be an assumption in the available, mostly physiological and survey-based research, that ASMR can be defined by a checklist of sensory experiences, yet gaps abound concerning how viewers themselves define their experiences. Due to the scarcity of research, particularly qualitative, this exploratory study seeks to address an identified gap. The aim of this qualitative study is to investigate how the ASMR consumer defines and engages with ASMR content, and what drives their consumption.

## Method

### Ethics

Ethical approval was obtained from the HOST UNIVERSITY'S Department of Psychology, Social Work, and Allied Health Sciences ethical committee. Ethical measures to address informed consent, possible participant distress, anonymity, and confidentiality were devised and implemented in line with the British Psychological Society's code of ethics and conduct (BPS) [76].

### Design

A qualitative approach was used to investigate ASMR from the perspective of the consumer. Exploratory qualitative methodology is inductive and therefore well suited to this under-researched phenomenon [77]. Data collection took the form of semi-structured interviews, which were conducted in part online to facilitate transnational data collection, as well as in person where possible.

### Recruitment strategy

Participants were recruited through a mixture of snowball and opportunity sampling. The participants were sought through word of mouth at HOST UNIVERSITY and through YouTube

video comment sections. The "word of mouth" approach involved informal discussions with classmates and making use of various messaging groups and discussion boards that were set up by course leaders to facilitate recruitment for student research projects. The information sheet was published in the comment section of thirteen ASMR videos on YouTube along with the researcher's university email as contact information. The videos selected included various forms of ASMR content such as roleplay, massage sounds, slime compilations, hand movements, mouth sounds and eating videos in attempts to capture an approximate gamut of ASMR content. Generic search terms such as "ASMR + keyword" were used to find this pool of videos. The search results were then influenced by the YouTube algorithm which highlights videos based on performance and personalisation. The videos also featured both male and female ASMR artists from various countries to avoid overrepresentation of one form of content, viewer, or gender-sex.

## Participants

A small sample size was used to allow for adequate analysis across the dataset [78] and was guided by the recommendations of Clarke et al. [79]. The first six individuals who contacted the researcher were selected for the interviews as more people volunteered (25 total) than could be accommodated in the time frame and scale of the study. As stated in recruitment materials, participants were selected on a 'first come, first served' basis to avoid selective sampling. Participants who responded after recruitment had been completed were informed of this and thanked for their offering to take part. Six participants were recruited in total with a gender distribution of three male and three female participants. The participants were located across five countries (Australia, Panama, England, Scotland, and Sweden) and ranged in age from 21–25 years old. Inclusion criteria required self-identification as an ASMR consumer, fluency in English, and being over the age of 18. Participants did not receive any material rewards for participation.

## Materials and procedure

Data was collected between January 2020 and March 2020. The literature indicates that ASMR consumption is predominantly a solitary activity [3] and it was therefore anticipated that the participants may not have self-reflected on, or spoken about, this experience prior to the interview. Taking this into consideration, it was anticipated that some guidance from the interviewer may be beneficial to assist the participant in organising their thoughts or exploring various concepts in greater depth. Semi-structured interviews were therefore chosen purposefully as they allow for the interviewer to gently guide a discussion while also promoting participant agency and direction [80, 81].

Interview appointments were decided collaboratively with each participant and held in a meeting room at HOST INSTITUTION if the participant lived in CITY. Participants abroad were allotted interview times based on availability and accommodating time differences. Remote interviews were conducted over Skype from the same meeting room that was used for in-person interviews. Participants were given information sheets either physically or digitally by email. Consent was obtained in writing before the interviews held in person and verbal consent was provided by remote interviewees. Unique identifiers were assigned to each participant to protect their anonymity in the transcripts. Once informed consent was provided, the semi-structured interviews were conducted, in general accordance with the pre-determined interview schedule. None of the questions were compulsory and the participants were encouraged to express to the degree they felt comfortable throughout the interview. Interviews lasted between 31 and 47 minutes with an average length of 41 minutes; upon completion of the

interview(s), interviewees were thanked for their participation. They were then provided debrief sheets (delineating the full purpose of the study, how the data would be used, and providing the contact details of the first author and their supervisor) which they were encouraged to keep, should they wish to later contact the interviewer. Remote participants were provided with this information by email. All audio files were uploaded to a Google Drive and interviews were subsequently transcribed verbatim in Microsoft Word. All interviews and transcripts were stored in an encrypted Google Drive and were only accessible to the authors. Anonymised transcripts are available via the Open Science Framework.

## Data analysis

Data was analysed using thematic analysis (TA) as described by Braun and Clarke [82, 83]. TA involves the identification of themes and patterns within the data [82, 84] and marries well to subject matter that pertains to lived experiences, personal perceptions, and opinions [82]. Theoretically we used a critical realist or contextualist framework [85]. This approach recognises the impossibility of direct access to an objective reality and instead considers the inherent subjectiveness of people's lived experiences and dictates that research enables interpretations of those experiences [86]. TA as an inductive strategy is utilised to identify, analyse, and report patterns within the data [82]. It does not need to rely on a particular epistemological stance or theoretical viewpoint. It is compatible with a constructionist paradigm making it a good fit for the current research; and in this case the TA was inductive rather than theoretical, the epistemology being from an interpretive approach, to understand the experiences of each participant [87].

The transcripts were read manually and iteratively to establish familiarity, looking for patterns and themes, in line with the aim/research question. For familiarisation purposes, the lead researcher took an idiographic approach to each transcript initially. Thereafter, codes were generated across the entire dataset (see Table 1). A colour-code was also utilised at this time to enable developing themes to be identified visually across all interviews [88].

Codes were then collated, and preliminary themes were created. Initial themes were then formulated, reviewed, and added to or modified based on constant comparison to the original text. No topics were excluded throughout the course of the analysis, instead themes were broadened to accommodate a range of reported elements of experience in the same theme. Weighting of topics was conducted according to emotional salience as dictated by the participant as well as the recurring emergent similarities across transcripts as observed through reflective analysis. Themes were then mapped in relation to the codes and the wider dataset to promote an accurate and holistic representation of the data. After mapping and defining the final themes, they were reviewed by the research team to adjust or edit to seek consensus that the themes chosen accurately reflected the data. This review contributes to inter-rater reliability and increases trustworthiness in qualitative research [89].

**Table 1. Example of colour-coding data for collation into themes, applied.**

| Data extract | Coded for |
|---|---|
| *. . . even if people have different feelings about what ASMR is I think having an idea of what other people think it is makes it easier for yourself to identify what is happening* | Who and what defines ASMR? (blue) |
| *. . . when there's two people on the same video I don't like it as much as when there's one, it becomes more of an intimate thing between people* | "Real" intimacy tailored to me (orange) |
| *. . .last year was probably the most stressful year I have ever gone through so I just needed things to take my mind off things so I had every relaxation app under the sun all downloaded on my phone but ASMR was the thing that helped the most because I feel like it's the easiest* | Relief on my terms (green) |

### Researcher reflexivity

Reflexivity is the greatest ally of the qualitative researcher and is the ability to not only think but rethink what effect the researcher has on the research process and how representations of others are constituted [90]. The lead author does not personally experience or consume ASMR content which could mean that they are more likely to interpret the findings from an outsider's rather than insider's position. Prior to initiating this research, the lead author watched a range of ASMR videos over a span of three months to get an idea of the content, community, context, and development of this genre of videos. Given their understanding of the ASMR scene, they endeavoured to the best of their ability to capture the voices of the participants, often stepping back to scrutinise their own positionality. As a 23-year-old woman, they were interviewing their own peer-group which could of course influence the nature of the interviews and discussions in ways untold. The double-hermeneutic impact of this research was also considered; not only what effect the researcher had on the participants but how the participants influenced the researcher [91, 92].

## Findings

The robust thematic analysis undertaken produced the following three themes: *Who and what defines ASMR?*, *"real" intimacy tailored to me* and *emotional relief on my terms.* The theme of *Who and what defines ASMR?* encapsulates the variety of what was classed as ASMR by the participants. The theme of *"real" intimacy tailored to me* takes into account that all participants felt they were able to find a tailored, personal, comfortably intimate experience within the ASMR domain. The theme of *emotional relief on my terms* synthesises participant disclosures that ASMR can be effective as a tool for self-soothing. For reasons of clarity, the themes are presented separately although they all play into the consumer's perspective of ASMR.

### Theme 1: Who and what defines ASMR?

The following quotes elucidate how even the most fundamental features such as what constitutes ASMR, and whether the gender or skill level of the artist matters, are up for discussion. Despite significant fluctuations in what counts as ASMR, all participants expressed an appreciation for the acronym, though few knew what it stands for. All participants were in agreement that ASMR can only be enjoyed alone.

All participants reported experiencing ASMR, though the content that they consumed to produce the response varied substantially:

> . . . *I feel like I definitely started off quite light, watching like soap cutting videos, like tapping noises and stuff but now it's grown into like literally anything like I feel like I can watch someone pretend to give me a doctor's appointment and it gives me the same relaxation as someone like pretending to brush my hair, so yea it's quite varied.*

> P1 21 year-old Male

> *My favourite one is this guy who is in the Australian wilderness and he just like builds stuff with his hands like mud huts. . . and you kind of see him, he never talks and that is so- that is my favourite by far.*

> P4 25 year-old Male

> *I like it when people just do rambles just talking about their day or whatever. . .*

P2 21 year-old Female

Intentional ASMR for the most part centres around an ASMRtist who addresses the viewer directly, however, the interviews revealed that even speaking was not a given in the triggering of the ASMR:

*I will instantly click off them if someone is speaking.*

P6 23 year-old Female

*Speaking, I think that's probably the most important thing. . .*

P2 21 year-old Female

When speaking was appreciated, the importance of understanding what is said, also varied:

*I've got an Italian girl I'm subscribed to and I have no idea what she's saying but she's just so good at it.*

P1 21 year-old Male

*Um I have watched some before in other languages, but I think it's probably more effective and I enjoy it a lot more in English.*

P2 21 year-old Female

*. . .I like to know what people are saying.*

P4 25 year-old Male

Watching patterns were diverse among the participants. Some reported watching and re-watching the same video which triggers ASMR for them. Others relayed a need to constantly experiment by watching different videos to achieve the desired sensation. The amount of actively engaged senses necessary to prompt the sensation also varied, where some reported to listening while others reported needing visuals as well:

*The key is variety, you get used to a certain trigger of ASMR and then once that happens you need to find something else.*

P1 21 year-old Male

*I know that I am that way, once I find a video i'll watch it like a million times.*

P4 25 year-old Male

*I used to be able to like watch it to get to sleep but I would like put my earphones in and shut my eyes and now it doesn't have the same effect I need to watch it and hear it.*

P1 21 year-old Male

*. . . almost every time fall asleep or I use it for relaxing and I just close my eyes I just put my phone down in my bed and I don't see what is going on.*

P5 23 year-old Female

When asked about the gender of the ASMRtist, participants had strong viewpoints. Some had a definite preference for female ASMRtists while others insisted the gender did not play a role in the videos or their ability to trigger the response.

*It needs to be a woman because I only have heard one man and I didn't, I didn't really like how he talked...*

P5 23 year-old Female

*...I like women with a sweet voice, like, so, not sounding like masculine but feminine, really feminine*

P5 23 year-old Female

*Males aren't very relaxing. It's all females that I usually watch.*

P2 21 year-old Female

*No, not at all no. Male and female, it's still the same thing.*

P1 21 year-old Male

*...it doesn't really matter the gender, if you've got a nice voice then it's nice to listen to.*

P6 23 year-old Female

Most participants reported watching ASMR on a near daily basis, despite this however, none of them were able to accurately recite the meaning of the acronym. All participants maintained that the existence of the term was important:

*...when you put something into words it makes it more real*

P1 21 year-old Male

*I think having an idea of what other people think it is makes it easier for yourself to identify what is happening...*

P4 25 year-old Male

*...if it didn't have an actual name it would kind of be like well what's this weird genre on YouTube?...*

P2 21 year-old Female

*...I was like oh my god I feel that...I was like oh my god I've literally felt that like my whole life and I never known it had like a definition...*

P1 21 year-old Male

From the genesis of the ASMR community, consumers of the phenomenon have contributed to its definition and maintenance by implicit and explicit boundary work. This was seen throughout the interviews by delineating ASMR from other auditory experiences as well as classifying some ASMRtists as 'professional' and more 'skilled' while others were considered 'amateur':

> *. . .I found Gibi* [a YouTuber] *and she was just really professional at it. . .*

P3 22 year-old Male

> *It depends as well on the quality of the ASMR artist. . .*

P2 21 year-old Female

> *I will also sometimes. . . find old videos better just because of how amateur they were.*

P3 22 year-old Male

> *I'd say experience definitely matters like there's people who have had (YouTube) channels for like years who clearly know what they're doing and then there are people who think they can just start a channel and just tap things.*

P1 21 year-old Male

The participants were purposefully recruited as they self-reportedly experience ASMR, though many participants maintained they do not, and never have, experienced tingling. This suggests that there are several diverse ways that the ASMR experience can manifest:

> *I don't think it's a full body experience, it's definitely all in your head it's like a brain thing.*

P1 21 year-old Male

> *I don't know if I really get like proper tingles the way some people talk about it because I just like feel relaxed and like tired from it.*

P2 21 year-old Female

> *Nah there is definitely no physical feeling to it I think its more a mental thing . . . as soon as you put it on its just like a placebo. . .*

P3 22 year-old Male

### Theme 2: "Real" intimacy tailored to me

The sexual or pornographic perception of ASMR was raised by participants in every interview, despite the absence of a dedicated question related to it. Elements of ASMR, such as whispering while making eye contact were considered either intimate or pornographic depending on individual interpretation. The intimacy portrayed in the videos was argued to range from maternal to platonic to sexual. Although opinions differed on what the intimacy portrayed in the videos represented, all participants felt they were able to find a comfortably intimate experience within ASMR, thus, the *tailored* aspect of the experience. Furthermore, the ASMR videos consumed by the participants were consistently tied to mostly positive memories. The nostalgia evoked often pertained to childhood, romantic partners, or films.

An important factor of this intimacy was found to be personal attention. The perceived exclusivity of the connection between ASMRtist and viewer was also highlighted:

> *. . . the personal attention, the taking care of you and a lot of it feels like, well it doesn't feel like, but they act like they're playing with your hair and stuff. . .*

P2 21 year-old Female

*. . . I like most this feeling of personal attention.*

P5 23 year-old Female

*. . . there's a common theme of like it's all about you so it's very personal . . . if someone was given more attention in the day maybe they wouldn't like it as much because they wouldn't need it. . .*

P3 22 year-old Male

*. . .the common theme is just like personal attention, it's something about you and not some-one else really. . .When there's two people on the same video I don't like it as much. . . when there's one it becomes more of an intimate thing between people.*

P3 22 year-old Male

*I feel like there's someone who understands how I feel and like a friend who told me what I need to hear. . . it's like a person for you.*

P5 23 year-old Female

The features of intimacy of the videos carried different meanings to different participants. Gender roles were apparent in the discussion of the female ASMRtist in terms of caregiving and the maternal bond. The line between sensuality and sexuality is also discussed:

*. . . when people like when they wanna feel safe or whatever they go to their mom and that's like a natural thing . . . maybe its just part of our nature to find that relaxing because you would feel safe by human nature. . .*

P3 22 year-old Male

*I think that the way a person speaks says (a lot) about how he or she acts. . . girls who speak that way are like more cute, more sweet with the people, say better things they're very feminine like a mother. . .*

P5 23 year-old Female

*. . . I would say there is something sexual to it I mean it's not like it replaces like um like por-nography or anything like that, but I think it's like something kind of similar*

P3 22 year-old Male

*. . . with the intentional ASMR its often like you know attractive women whispering into a mic that's what I don't like about it, that's what makes it so you know creepy or like it adds this sexual thing to it. . .*

P6 23 year-old Female

Beyond personal perceptions, the participants also expressed their perceived public percep-tion of the sexual nature of ASMR:

*. . . I think generally people only know that side of it would probably just think oh it's a bunch of perverts who are listening to weird shit.*

P6 23 year-old Female

*I think there is this sort of undertones of like my friends who say it's weird I think they might think that its maybe perhaps a bit sexual. . .*

P1 21 year-old Male

*I think some people think it's like slightly pornographic and like sexual . . . I think it's incorrect that perception.*

P2 21 year-old Female

Features of socially-intimate encounters were prominent in the interviews, particularly authenticity of connection and proximity.

*I think it's proximity. . . it's just someone being close to you.*

P1 21 year-old Male

*Yeah I think so but then also it's a bit off putting when they're like really close like I don't like it when they are like right in the camera.*

P2 21 year-old Female

*It's not quite as relaxing if they are sat quite far away from the camera.*

P2 21 year-old Female

Proximity also manifested in the preference of the smartphone for technological device used to consumer ASMR content.

*I do only watch it on my phone. I would never think of watching it on a tablet.*

P6 23 year-old Female

*. . . it's easier on my phone. . . I'll switch my TV off and put my laptop away and I'll just have it (ASMR) up.*

P1 21 year-old Male

Authenticity manifested mainly in a distaste for roleplay and overt acting in the ASMR videos. Many of the comments reflect a desire for genuineness from the ASMRtist:

*I don't really enjoy the roleplay so much, not when they like dress up and don't look like themselves . . .*

P3 22 year-old Male

*I think if they try too hard it can like get, it can ruin the effect. . .*

P2 21 year-old Female

*I would probably like it if they were doing the exact same stuff but were not in costume.*

P1 21 year-old Male

*I watched it (roleplay) because it's the most common content that ASMRtists upload and I don't like it because it feels like false. . .*

P5 23 year-old Female

Plausibility was an additional factor involved in ASMR videos which was highlighted by the participants. This ranged from appreciating videos with more realistic situations to videos which mirrored the lived experience of the viewer:

*. . .it's (roleplay) not real I don't like that way I like more when it's something that you can have in front of you like a friend telling you everything is going to be okay but if it's saying "I'm an alien and I'm going to save you" it sounds I don't know, stupid. . . that's not true and it will never happen.*

P5 23 year-old Female

The tailored aspect of the intimacy was reflected in how each participant linked the ASMR they watch with a unique positive memory from their own life:

*The first time I experienced ASMR wasn't on YouTube it was with voice notes from WhatsApp from a classmate . . . she speaks sooo slow, she had everything; the slowly-ness, it was a girl, she was sweet and I was listening to her voice notes when I was feeling anxious and lost.*

P5 23 year-old Female

*. . . and you'd know it was cold outside and you were in your warm bed and there's sort of rain going on outside and the sort of sound of people's fingernails and things just anything that has a kind of like rain-esque quality to it.*

P6 23 year-old Female

*I just thought back and realised that one time I felt it (ASMR) was with an ex and we were reading books and she started reading the book that she was reading to me and then I got those feelings. . .*

P4 25 year-old Male

*The first time I experienced ASMR it was a long time ago. . . there is a scene in that movie that I got super ASMR-y as a kid . . . recently when I discovered ASMR like that clip was on the ASMR subreddit. . .and I was like oh shit that's what I was feeling and I remember loving that scene as a kid . . . that feeling has always been there I just haven't like sought it in the same way.*

P4 25 year-old Male

*I've always felt it (ASMR) as a child but like watching YouTube videos it did take a while for it to like kick in.*

P1 21 year-old Male

*I used to work in a coffee shop um like you'd have to pour the coffee beans. . . the packets that had the beans in. . . made a really nice sticky noise almost so I actually experienced it once then. . .*

P2 21 year-old Female

## Theme 3: Emotional relief on my terms

Most participants cited a pre-existing struggle with mental health as what led them to discover and continue to engage with ASMR. This manifested as disordered eating, anxiety, panic attacks, depression, and/or loneliness. In this way ASMR can be seen as a tool for self-soothing.

*I watch it when, more when I feel anxious. . . mostly in the night when I am just thinking about everything before sleeping.*

P5 23 year-old Female

*. . . it was quite a tumultuous time and I felt my mental health took quite a dip and I was looking at things I could do just to like make myself feel better and ASMR was the thing that came up. . . So if I ever felt stressed I would just like take a wee minute away from like uni work thinking and just my put my earphones in and sit for a while.*

P1 21 year-old Male

*I had a panic attack one time and it (ASMR) was the exact opposite of a panic attack. . . it's like being washed over with relaxation. . .*

P4 25 year-old Male

Watching ASMR videos was likened to accessing a meditative state by many of the participants. The soothing element was evident in ASMR's reported ability to change the course of thought patterns or to temporarily stop thinking entirely.

*If it's silent nowadays it feels almost weird like I'm just so used to putting it on so I don't know it makes me feel calm and it just gives me something to focus on.*

P3 22 year-old Male

*I think for me it's literally like earphones in, put a video on, switch off.*

P1 21 year-old Male

*I would like to get into mindfulness, but I feel like I get a similar experience from ASMR.*

P2 21 year-old Female

*. . . as you put it on its just like a placebo because you know it and you're expecting it and then as soon as it goes on you're just like ready to calm down.*

P3 22 year-old Male

It was also observed that ASMR was perceived to be 'easier' than meditation in that it did not ask anything of the individual on the receiving end.

*. . .I had every relaxation app under the sun all downloaded on my phone but ASMR was the thing that helped the most because I feel like it's the easiest.*

P1 21 year-old Male

*Yeah I mean it's like guided meditation but you're not being guided you're just watching something and letting it happen to you.*

P4 25 year-old Male

*. . . it's like meditation but I can be watching something instead of just like closing my eyes and thinking of nothing. . .*

P4 25 year-old Male

## Discussion

This study aimed to investigate what ASMR *is* from the perspective of the consumer. Previous research [36, 93] has focused mainly on validating the response or exploring what enables it, implicitly under the assumption that ASMR is one uniform and definitive phenomenon. The results of the current study address an identified gap in the literature and add to the current state of knowledge regarding ASMR. They also provide a different perspective from previous findings, particularly regarding how ASMR manifests for the individual(s). The established physiological basis of ASMR [3, 24, 93] was generally not reflected by the participants' responses. While few participants reported an experience of tingling, most believed it was more of a psychological effect that did not reflect real somatic sensation.

Some participants reported music as having the ability to spur an ASMR reaction, while other participants found that certain sounds which for others produce ASMR personally provoke a feeling of disgust to them. These findings contribute to the researched similarities between ASMR and other sound-based conditions such as frisson and misophonia [51, 57, 94]. These findings underscore the individuality and variety which was found in eliciting ASMR. A thematic analysis determined three themes: *who and what defines ASMR* which identified the variety of what was classed as ASMR and what content was consumed to produce the response; *"real" intimacy tailored to me* reflected the idiosyncratic perception of intimacy made possible through ASMR; and *emotional relief on my terms* which reflected the use of ASMR as a tool to self-soothe. Cumulatively, these themes create an overarching conceptualisation of ASMR as a personalised tool for providing comfort.

The qualitative and exploratory nature of this study uncovered gaps in the current academic perception of ASMR. A large portion of the available research which preceded this study did not fully account for the vast range and robust individuality of each of the four factors: the motivations behind consuming ASMR content, the ideal experiential conditions necessary to maximise its positive effects, the type of content preferred and the perceived effects of consumption. Nevertheless, ASMR's soothing ability formed a common ground shared by all participants. Emotional comfort and a desire for authenticity was present throughout the analysis and in all themes. Authenticity of the ASMRtist was found to be an important element in portraying a sense of intimacy to the consumer and in evoking nostalgia. The presence of nostalgia was noted in the construction and content of the ASMR videos and possibly also as an experiential feature which heightened the response. Consuming ASMR was found to unanimously provide a positive experience for the participants, many of whom were struggling, or had struggled, with mental health issues. While traditional meditation may ask an individual to

sacrifice entertainment in exchange for a possibly uncomfortable confrontation with their own thoughts, ASMR places no such requests. The current study found it to be a shortcut to achieving comfort, a feeling of personal attention and a calmer mental state.

## Who and what defines ASMR?

Several participants reported not feeling tingles or anything remotely physiological during their ASMR consumption but cited that they still felt that they experienced ASMR. The type of content and experiential conditions necessary to produce the response varied greatly. Neither of these findings are accounted for in the current definition of ASMR but are increasingly common in the existing literature [7]. Lochte et al. [24] found a tingling sensation only comprised a minority (5.6%) of an ASMR testing session whereas relaxation accounted for over half (54%). This is consistent with the current findings where participants cited ASMR's ability to induce feelings of relaxation but, as aforementioned, contrasts with the original definition of the phenomena. This is important to note as, in the case of research, a definition is necessary in providing common ground [95, 96]. Previous research has referred to individuals as 'having' ASMR [17, 46], however, the current findings question what exactly *having* ASMR means. Although defining ASMR has been a focus in previous studies [3], for consumers, there may be some strength in a more flexible definition or no single definition at all. Our interview data suggests that the ambiguity of what constitutes ASMR lends itself to many definitions. The demarcation of ASMR as constructed by the individual was specific to their experience. A participant who struggled with disordered eating cited ASMR featuring chewing sounds as satisfying her in a way which made her feel less hungry. Other participants cited ASMR as having the ability to reduce feelings of loneliness by providing an authentic connection.

None of the participants in the current study were able to accurately recite the full form of the acronym, though the consensus was that the sensation did need a definition. This assertion is in line with one of the most impactful facets of the larger ASMR community: their engagement with boundary work, consciously and explicitly defining what ASMR is and is not [61]. It was also acknowledged by the participants that, despite not knowing what ASMR stands for, the mere existence of the term makes the experience feel more 'real' and/or legitimised. In this way, it is argued that ASMR is more of a semantic and symbolic term than a pragmatic one [97].

A further example of how the participants define ASMR was in their identification of the skill level of the ASMRtist. To assert that the ASMRtist is skilled indicates that what they are doing could somehow be quantified and that they could also be bad at it. Participants outlined a scope of ability in triggering ASMR ranging between amateur to professional. Most preferred a 'professional ASMRtist'; besides time spent creating ASMR content, it was not clear exactly what constitutes *professional*. There may also be a self-imposed impression of professionalism to attempt to bring validity to the practice [98–100]. As with the acronym, the professionalism expressed when referring to certain ASMR creators may have been manufactured in attempts to make ASMR feel more tangible and thereby improving its effects. This action can be compared to the construction of meaning in online gaming. Juul [101] found that online gamers were rule-abiding even when the rules were optional, as without the limitations and affordances provided by rules the game lost meaning and was no longer entertaining. This could explain why our participants endeavour to define the abstract practice of ASMR in terms of skill. It may heighten the ASMR response to differentiate between a professional video and an amateur one, thus self-imposing rules to seemingly enrich the quality of the experience.

## "Real" intimacy tailored to me

Intimacy can be defined as a dyadic exchange involving a physical, emotional, and cognitive component mainly rooted in the shared experience of what is personal [102]. The resultant feelings of intimacy reported by the participants were said to be related to a variety of factors including maternal caretaking, platonic closeness, and provocative sexuality. Notably, female ASMRtists were preferred for their 'maternal' and 'feminine' disposition yet they were also pinpointed as being the reason that ASMR is seen as sexual. Although the novelty of ASMR content is often underscored, the dichotomy of women as maternal figures or sexualised objects that is present in traditional media seems to exist in parallel within ASMR [103, 104]. This ambiguous narrative of intimacy found in the current study is in line with work by Smith and Snider [61] who claim that ASMR uses social and intimate affective experiences such as whispering, personal addresses and eye contact to prompt a sensory experience. This response is then curated and maintained through technology. From this perspective, it can be argued that ASMR is not a distinct sensation; it is merely a digitally mediated indulgence of a natural human response [105–107]. The social interactive nature of ASMR is underscored by an analysis on the role of the viewer in ASMR consumption [108]. Bennett [109] argues that the effects of ASMR are largely social and are related to recognition.

A common theme in ASMRtists' discourse is addressing the viewer personally. For example, in doctor-patient roleplays, the ASMRtist will often say something about 'you', *the* patient, not something about *a* patient. Bennett [109] contends that addressing the viewer directly taps into reflexive social mechanisms. Despite the pre-recorded and impersonal nature of the content, the viewer's cognitive and physiological reaction to the material could mimic that of a genuine interpersonal address. Lochte et al. [24] strengthened this argument by identifying activations in regions of the brain associated with socialising during the experience of ASMR. From how women are perceived in the videos to the nature of the response, there may be merit to the argument that ASMR is merely a novel amalgamation of pre-existing concepts.

ASMR video construction and consumer habits mirrored several face-to-face social customs despite the digitally mediated nature of the content. Propinquity was a particular focus, both in terms of the consumers' proximity to their screen and the producer's proximity to the camera. Having the ASMR creator closer to the camera was emphasised as important by several participants. This is further supported by a study which showed a consistent desire for the ASMRtist to appear closer on the screen than further away [93].

A different—but related—explanation for this preference could be found in the form of a pre-existing smartphone reliance. Using the recently developed, albeit heavily criticised [110], smartphone addiction scale [111], Bian and Leung [112] found that the most common catalyst for developing so-called smartphone addiction was loneliness. Beyond 'addiction', it has been said that our smartphones have become our most intimate partner [113]. Loneliness was a recurring theme throughout most interviews. It manifested as an incentive to discover ASMR and to use it as a tool to stave off feelings of isolation and seclusion. Loneliness may therefore be the cause behind both consuming ASMR and intensively using one's smartphone, activities that occasionally coincide. From this perspective it could also be argued that ASMR exists due to its unique position in contemporary sociocultural and technological development.

The plausibility of the scenario being enacted by, and the perceived authenticity of, the ASMRtist were found to be two of the most important factors for the participants. It is posited that the more authentic and plausible the ASMR content, the more likely it is to spur a positive memory consequently tapping into the positive affect associated with nostalgia [114]. The term *algia* denotes the longing for nostos, meaning home [115]. Linguistically, the term demonstrates a desire for returning to the home, a symbolic journey to a place of comfort [116]. As

nostalgia has been found to conjure mostly positive affect [108], it is conceivable that ASMR harnesses this effect through both producer and consumer intentions and actions. This could manifest as the producer drawing on generic positive experiences such as care-taking and personal attention. While the viewer selectively (consciously or unconsciously) recalls positive memories to enhance rather than diminish their experience of ASMR. Evidence for this mechanism can be found in the nature of the numerous memories recalled by the participants in relation to ASMR; being read aloud to by a romantic partner, being in one's warm bed as a child and other memories associated with comfort. The participants also expressed a feeling of overproduction in recent ASMR videos and shared that often they preferred lo-fi sound, a purposely lower, more imperfect sound quality. Along with this, an aversion for roleplay was articulated. The combination of preferring a plausible scenario depicting a real person coupled with lo-fi sound could be said to create a pseudo memory; something which feels like it could have happened and is constructed to feel familiar and old thus generating nostalgia. Research has related this deliberate indulgence in nostalgia to a desire to slow down rapidly developing technology by escaping to 'simpler' times and thereby reducing the cognitive load [117]. The experience of nostalgia has also been found to counteract boredom, anxiety, and loneliness [114]–three states which were also credited to be soothed by ASMR. One study found participants experiencing nostalgia were able to tolerate noxious cold suggesting a physiological component of the experience [16] a further similarity between ASMR and nostalgia. As outlined above, this analysis found several similarities between nostalgia and ASMR suggesting that ASMR consumption may involve an interaction of these concepts.

## Emotional relief on my terms

Almost all participants mentioned a pre-existing mental health struggle as the catalyst for discovering and continuing to engage with ASMR content. Despite different pathologies, all appeared to prompt the individual to seek relief from discomfort in the form of ASMR. The comforting ability of ASMR could be afforded by its conscious promotion of positive affect [14].

All participants consumed ASMR using the internet, chiefly through YouTube and Instagram. It is therefore argued that engaging with ASMR content could be an example of the online disinhibition effect [118] where social inhibitions are slackened in cyber environments. Although this can lead to recklessness and a lack of personal accountability [119], it can also create a comfortable and relaxed state without fear of embarrassment [120]. This effect has been shown through participation in online roleplay where anonymity was a predictor of player engagement due to low emotional risk [121]. Similarly, the anonymous and one-sided nature of ASMR videos allows for a unique connection without jeopardising any emotional investment. In contrast to face-to-face interaction, where interpersonal conflict is possible, in ASMR, the viewer can seemingly reap the benefits of an intimate relation without any emotional or physical exertion. The desire for a solitary viewing environment when consuming ASMR could also magnify these effects by removing any sources of external observation. A study by Markus [122] found that even with no clear evaluation criteria, participants performed differently when in the presence of others. This finding has been demonstrated numerous times [123–125]. This further justifies the desire for solitude to promote and enhance relaxation through magnifying certain positive social customs such as verbal affirmation [126] and eye contact [127, 128] while stripping away potentially negative external influences. The presented lopsided digitised exchange creates a unique socially affirming practice in a solitary viewing environment.

Several participants equated ASMR to meditation and most agreed that ASMR was simpler and more reliable in its ability to encourage relaxation. The perceived ease of ASMR may be

supported by its auditory focus. In a study investigating the sensory requirements for an effective meditation application, it was found that, despite the significant role that visuals and touch play, audio was the most important feature in relieving stress [129]. Many participants reported not watching the ASMR videos at all but instead solely listening to the audio, especially prior to sleep. This finding is consistent with the existing literature where relaxation and sleep are the two most common goals cited for engaging with ASMR content [41, 130]. Despite some similarities, meditation and ASMR have very different goals. Meditation aims to observe personal thought patterns and increase awareness to promote a greater sense of peace for the individual [131–133]. Most participants in the current study, however, aimed to ignore their own thoughts and engage in a type of active distraction. This was also evident from the viewing patterns, where all participants reported consuming ASMR at night before falling asleep. While meditation encourages conscious presence in each moment, the findings of the current study demonstrate the tendency to use ASMR as a distraction instead of dealing with whatever problem directly. The theme title '*on my terms*' stems from this observation of self- soothing without necessarily addressing the issue. This deviation from conscious processing sets ASMR apart from meditation though they are considered similar by participants due to the calming aspect of both.

## Strengths, limitations & possible implications

A key strength of this study was in its novel approach in terms of methodology, particularly recruitment. Few studies have been conducted on ASMR and fewer still that are qualitative. Additionally, the sampling process whereby active consumers of ASMR content rather than 'passive experiencers' of the perceived physiological response were recruited; by this it is meant that the current study sought to highlight those who engage with ASMR content rather than sampling for those who merely possess the ability to experience it. The implications of this choice in sampling could entail discovering more diverse uses of ASMR as well. A further strength of this study was the use of semi-structured interviews with open-ended questions. Allowing each participant to steer the interview facilitated the exploratory nature of the study and the open-ended questions aided in decreasing bias by not suggesting answers within the questions or otherwise directing the response [82].

A limitation of the current study is the method of advertisement used to enlist participants online. Although a concerted effort was made to advertise under a range of different ASMR videos, there is a risk of low transferability of the current findings as they may mostly reflect a narrow pool of ASMR consumers based on the YouTube comment sections where the study was advertised. An improved version of this sampling tactic could be recruiting a larger sample of participants. This sampling strategy may have a better chance of ensuring less sampling bias than the current approach. Moreover, the individuals who are willing to partake in interviews regarding ASMR may not be representative of the ASMR population at large [134, 135]. Certain individuals who volunteer to be interviewed may be "stand outs" rather than prototypical representatives of the ASMR community.

Furthermore, the study lacked a uniform interview setting. Some participants attended in person while others did so from home over Skype. Interview behaviour has been shown to vary depending on online or offline setting [136, 137] hence it is conceivable that mixing these methods of data collection could have impacted the findings. A form of social desirability bias may have also been present in the participants, creating an additional limiting factor of the research [138]. The non-naturalistic interview setting [139] in conjunction with the mix of online and offline interviews may be a limitation of the current study.

### Future research and practical applications

Future research should seek to explore cultural differences in ASMR viewing behaviour / consumption. Most studies, including this one, have considered ASMR more broadly as a global concept. As this phenomenon becomes better understood, exploring potential local resonances could yield more precise insights into consumer behaviours and motivations.

Bringing ASMR to the mainstream with the help of research could help it to become a viable option for self-soothing, for those outside of the ASMR community as well. Where other media such as films and music may produce an ASMR experience as a by-product [59], ASMR videos are unique in how their sole intention is to make the viewer feel good. This is an important finding to incentivise future research as ASMR could have large and far- reaching applications for mental health. This also comes at a time where there is an increasing interest in developing, and an increased demand for, online resources for mental health issues [140–142]. The COVID-19 pandemic has transformed this pre-existing demand for accessible online mental health services into a necessity through lockdowns and quarantines that have impeded on traditional care avenues. The rise of telehealth and other digital care solutions has recently become a top-level priority in several healthcare sectors further exemplifying this shift [143].

Participants' habitual consuming of ASMR media on their phone in bed, represents a practice which would not have been possible pre-smartphone or pre-internet [144]. For future research, it could be illuminating to further study the intentional consumption of ASMR as an emergent habit afforded by consumer technology as well what our device proximity preference means in terms of ASMR consumption.

The phrasing used to recruit participants is of further importance regarding the discourse involved in ASMR research. Sampling those who *experience* ASMR versus those who *consume* it could present two distinct groups. As ASMR has been positioned as something that an individual either can or cannot experience, opening the sample to those who consume but do not necessarily *experience* ASMR could change the sample entirely. For clarity and generalisation purposes, future research should explicitly state and justify what type of ASMR consumer was sampled. Previous research often lacks this distinction, and it is therefore argued that this could be a confound. The current study purposefully sampled for ASMR *consumers* as they make up the broader community. Sampling beyond those who claim to experience ASMR was also thought to reveal wider uses of ASMR beyond prompting tingles. Efforts to understand a more multidimensional experience of ASMR are however underway with more recent studies placing emphasis on altering of consciousness rather than physiological feedback [41]. Notwithstanding, this difference in sampling could also seek to explain why the results of the current study do not entirely reflect the bulk of the previous literature, and the present perception, of ASMR and its consumers.

The lack of defined boundaries in ASMR provides a customisable experience which has been shown to accommodate many different needs and desires. This is important to note for potential future useful applications of ASMR whereby defining it too strictly could prevent a wider audience from engaging with it. This is also important to note for future research purposes as a sampling consideration, how strictly the phenomenon is defined will impact the type of participants who engage.

## Conclusions

In conclusion, there is evidence to suggest that the ASMR experience may be an intensified version of an existing human response rather than a new phenomenon altogether [62, 72]. Rather than relating to the response, the novelty of ASMR may instead be found in the method of accessing the experience; namely through ASMR content. Both previous literature and

participants in the current study viewed defining ASMR as a useful and necessary pursuit, though reaching a consensus on a definition proves problematic.

Academia has also, to some extent, begun to explore the digitally mediated intimacy associated with ASMR [61]. This study furthered the investigation of technology as an enabling factor for this intimacy, concluding that the smartphone largely serves as a gateway to experiencing ASMR effectively.

In our present fast-paced society, consumer technology continues to advance rapidly. We engage in fewer face-to-face interactions [1] and the proportion of people with anxiety and depression continues to increase [145, 146]. ASMR has come to exist at the intersection of these new truths. The ASMR 'safe haven' in an otherwise chaotic and loud internet environment merits further exploration.

## Author Contributions

**Conceptualization:** Enya Autumn Trenholm-Jensen.

**Data curation:** Enya Autumn Trenholm-Jensen.

**Formal analysis:** Enya Autumn Trenholm-Jensen, Lindsey Burns.

**Investigation:** Jill Ellen Trenholm.

**Methodology:** Enya Autumn Trenholm-Jensen, Lindsey Burns, Christopher James Hand.

**Project administration:** Christopher James Hand.

**Supervision:** Jill Ellen Trenholm, Christopher James Hand.

**Validation:** Jill Ellen Trenholm, Christopher James Hand.

**Writing – original draft:** Enya Autumn Trenholm-Jensen, Lindsey Burns, Jill Ellen Trenholm, Christopher James Hand.

**Writing – review & editing:** Enya Autumn Trenholm-Jensen, Lindsey Burns, Christopher James Hand.

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
