## [Decision Letter · Decision Letter 0]

11 Aug 2022

PONE-D-22-10021Beyond tingles: an exploratory qualitative study on Autonomous Sensory Meridian Response (ASMR)PLOS ONE

Dear Dr. Hand,

Thank you for submitting your manuscript to PLOS ONE. After careful consideration, we feel that it has merit but does not fully meet PLOS ONE’s publication criteria as it currently stands. Therefore, we invite you to submit a revised version of the manuscript that addresses the points raised during the review process.

We look forward to receiving your revised manuscript.

Kind regards,

Jianhong Zhou

Staff Editor

PLOS ONE

" ext-link-type="uri" xlink:type="simple">https://journals.plos.org/plosone/s/file?id=ba62/PLOSOne_formatting_sample_title_authors_affiliations.pdf".

2. Peer review at PLOS ONE is not double-blinded (https://journals.plos.org/plosone/s/editorial-and-peer-review-process). For this reason, authors should include in the revised manuscript all the information removed for blind review.

Reviewers' comments:

Reviewer's Responses to Questions

**Comments to the Author**

1. Is the manuscript technically sound, and do the data support the conclusions?

Reviewer #1: Partly

2. Has the statistical analysis been performed appropriately and rigorously? 

Reviewer #1: N/A

3. Have the authors made all data underlying the findings in their manuscript fully available?

Reviewer #1: No

4. Is the manuscript presented in an intelligible fashion and written in standard English?

Reviewer #1: Yes

5. Review Comments to the Author

Reviewer #1: This is an interesting and well-written paper that provides a more nuanced understanding of the ASMR phenomenon as it is experientially understood and defined by the ‘ASMR Consumer’; thus, making for a potentially insightful contribution to the broader quantitative literature, and I thank the authors for their work.

The manuscript is largely well presented and researched. However, there are several areas I feel need be addressed to enhance its overall clarity and transparency for readers. I list these below:

Title:

A minor point, but ‘an exploratory qualitative study of the…’ may read better than ‘an exploratory qualitative study on…’. Just a suggestion.

CRediT Author Statement:

I believe author 1 should also be credited for Data Collection also.

Abstract:

If possible, try to include a summary of the three themes, either individually or together to be more concise, to help sell the paper and grab the reader’s attention. I appreciate the authors try this on lines 34-35, but I don’t feel this does justice to the three themes in capturing the take-home message i.e., the variability of the ASMR experience that includes shared elements of perceived ‘intimacy’ and psycho-therapeutic relief.

Keywords:

These appear in the manuscript submission table up top, but not in the actual text, below the abstract. Just double-check that they’re there.

Introduction:

This is largely well written and researched, however there are several instances where the text slips into past tense, e.g., lines 58-60 ‘as the focus of this paper was to…’. Try to keep the introduction in the present tense: ‘as the focus of this paper is to…’

You reference a scoping review, lines 85-87, is this published? If not I would rephrase this, for example: a cursory review of the literature suggests…, or simply state there is little qualitative literature on ASMR.

You reference Holster et al., lines 88-93, and credit using their framework to guide this investigation, however it is not clear how the framework guided the research and is not mentioned again. I would either make clear in the method, results, or discussion, how the framework informed the study, or remove mention of it as guiding the study.

There are several instances where papers are cited by author ‘and colleagues’, instead of author et al. Change to et al. for consistency throughout.

Line 112: delete ‘(2018)’- it is not needed here.

Line 117: referencing more than two papers at once e.g., [25,26,27] can be shortened to [25-27]. There are other instances where this happens in the manuscript so check for these also.

Lines 279-281: again, the writing slips into past tense here. Consider revising to: ‘Due to the scarcity of research, particularly qualitative, this study seeks to…’ Also, I’m not sure about the current aim as it is written, particularly the aim to identify ‘who these consumers are’- I don’t see this being addressed either in the findings or elsewhere, and what would it mean to identify them- in terms of age, sex, etc.? I feel the findings better address an aim to explore how the ASMR consumer defines and engages with ASMR content, and what motivates their consumption.

‘Materials Methods’:

This is the section I feel needs the most work to improve the paper’s overall clarity and transparency in its process.

First, I would revise the section title to ‘Method’. Also consider adopting the following structure and subheadings for this section: ‘Design’; ‘Materials’; ‘Recruitment Strategy’; ‘Participants’; ‘Procedure’; ‘Ethics’; and ‘Data Analysis’.

The ‘Design’ subheading will be particularly useful for the reader to make clear the rationale for using a qualitative approach and semi-structured interviews. Moreover, some of the existing writing in the manuscript could be better placed here, such as lines 313-324.

The recruitment strategy, lines 290-300, would be better presented with its own subheading for added clarity, and needs more specificity, particularly in terms of making clear the ‘word of mouth’ process. This is briefly stated with little detail about how word of mouth recruitment was conducted- how were potential participants identified and approached? This is crucial I feel for both research transparency and ethics.

Regarding participants, lines 301-310, again this would be better presented with its own subheading, and specifying the mean age and standard deviation appears unnecessary, especially for a sample size of 6 with little age range; simply stating the age range would be sufficient and in keeping with qualitative research generally.

Lines 302-305 also need more clarity. While I recognise practical constraints and need to limit recruitment to timescales etc., were participants told about the ‘first come, first serve’ recruitment, and were they told if they had not been selected to take part but still thanked for volunteering their time?

Ethics: Line 286, ‘No significant ethical implications were identified’ I think I understand the authors’ meaning, but I’m wary of the phrasing- almost all research carries potential ethical challenges etc. so I would avoid statements that could be misconstrued to mean otherwise. Better, I think, to delete this line or rephrase to clarify that no significant ethical concerns were raised during the conducting of this research.

Also on ethics, I would consider revising lines 286-287 to read: Ethical measures to address informed consent, possible participant distress, anonymity, and confidentiality were devised and implemented in line with the British Psychological Society’s ‘Code of Ethics and Conduct’.

Data Analysis: the authors provide a good epistemologically informed rationale for using thematic analysis, lines 347-358, however I would delete ‘without an explicit hypothesis’ (line 348).

That said, lines 359-370, could be better clarified to describe the analysis of data and construction of themes. Structuring this paragraph around Brain Clarke’s prescribed analytic steps, in terms of data familiarisation, generation of initial codes and themes, and defining of themes, would help make clear this analytical process, while also allowing the authors to specify their use of colour coding and review of themes within the research team. Also, lines 388-394 of the Findings should go here, along with ‘Table 1’ for the purposes of illustrating the analytical process- which is good to see.

Researcher Reflexivity: it is nice to see a reflexivity included- as they rarely are in published qualitative literature in my experience. Thank you.

Findings:

The findings largely read well, however there are a few points I think would help improve the overall presentation and clarity or the reporting.

As stated already, lines 388-394 and Table 1 should be moved to Data Analysis- only report the themes you identified in this section; not how these themes were constructed.

Also, it helps to briefly introduce the three themes at the start of findings section -after first naming them- to help orient the reader as to what the themes generally are, before then going into further detail to report each theme individually.

I also think it would help to specify the age and sex of the participants alongside their excerpts, to help the reader discern the participant’s voice; so, for example, going from ‘P1’ to ‘P1, 22-Year-Old Female’. Stylistically, I largely preference the use of pseudonyms to help in this also, however given the current reporting, keeping to ‘PX’ identifiers will suffice.

I think it would also help to have a brief concluding paragraph at the end of each of the theme sections that summarises and makes clear to the reader the central argument of the theme, and its elucidating of our understanding of the ASMR Consumer.

Lastly, the same participants are quoted multiple times to illustrate the same individual interpretative point. While not necessarily an issue, it can appear somewhat redundant. But optional if you want to address this.

Discussion:

Before comparing the findings to previous literature at the start of discussion, try to again provide a concise summary of the themes, to reinforce to the reader the central findings of the paper and what it adds to our understanding of the ASMR Consumer.

Also consider adding a dedicated subheading for ‘Future Research’ that lists your recommendations for future research, instead of having these sparsely spread across the discussion as they are currently. This is useful as quick reference for the reader.

Similarly, consider having a devoted subheading for ‘Practical Recommendations’, instead of merging this with the strengths and limitations section. Again, for ease of reference and clarity.

Strengths and Limitations: Lines 926-927: I don’t know if this claim to the recruitment of international participants as contributing to the understanding of ASMR globally is especially merited. While it speaks to the global viewership of ASMR, and is something that future research may seek to build on, in terms of understanding sociocultural variances and the ASMR consumer, this international aspect doesn’t do much to expand on what aspects of ASMR are appreciated globally here, based on my reading of the paper. That said, the recommendation for future research to pursue a culturally focused analytical lens to this issue is a good one.

Lines 933-938: I’m not sure I understand this point about the reliance on self-report and potential biasing impact and idealised need to measure and test the views shared. Isn’t the point of this research to understand the consumer’s own understanding and experience of ASMR? While it is possible the accounts shared may be impacted by unconscious processes -or even deception by the participants- it is hard to consider alternative ways of probing the consumer’s own thought and opinions, as stated in lines 937-938, and it is equally hard to consider how we would ‘test’ the accounts shared- be it via quantitative or by physiological measures—especially given the point of the paper being to recognise ASMR as more than a physiological ‘tingle’ in the eyes of the consumer.

Lines 939-950: the limitation concerning the transferability of findings is indeed a valid one and worth mentioning, however I do not believe that your proposed fix: to recruit a larger pool of potential participants and then select six at random, does anything to change or improve the transferability of findings- wouldn’t this or a ‘first come first serve’ approach to recruitment serve as an equally random and still non-representative approach to recruitment? I don’t see what would be gained by gathering a large pool of potential participants first, then selecting six at random.

Lastly, the ‘Conclusion’, lines 960-995, raises multiple questions, though I wonder if this section could be made a bit tighter? Often the conclusion is just that- a short concluding paragraph that summarises the findings, what they add to the literature, and how future research and practice may build on this. I would recommend trying to keep to this general structure and brevity to really ‘sell’ the take home message to the reader.

6. PLOS authors have the option to publish the peer review history of their article (what does this mean?). If published, this will include your full peer review and any attached files.

Reviewer #1: **Yes: **Benjamin Lond

---

## [Decision Letter · Decision Letter 1]

8 Nov 2022

Beyond tingles: an exploratory qualitative study of the Autonomous Sensory Meridian Response (ASMR)

PONE-D-22-10021R1

Dear Dr. Hand,

We’re pleased to inform you that your manuscript has been judged scientifically suitable for publication and will be formally accepted for publication once it meets all outstanding technical requirements.

Kind regards,

Jane Elizabeth Aspell, PhD

Academic Editor

PLOS ONE

Reviewers' comments:

Reviewer's Responses to Questions

**Comments to the Author**

1. If the authors have adequately addressed your comments raised in a previous round of review and you feel that this manuscript is now acceptable for publication, you may indicate that here to bypass the “Comments to the Author” section, enter your conflict of interest statement in the “Confidential to Editor” section, and submit your "Accept" recommendation.

Reviewer #1: All comments have been addressed

2. Is the manuscript technically sound, and do the data support the conclusions?

Reviewer #1: Yes

3. Has the statistical analysis been performed appropriately and rigorously? 

Reviewer #1: N/A

4. Have the authors made all data underlying the findings in their manuscript fully available?

Reviewer #1: Yes

5. Is the manuscript presented in an intelligible fashion and written in standard English?

Reviewer #1: Yes

6. Review Comments to the Author

Reviewer #1: As acknowledged in my first review, this is an interesting and well-written paper that provides a more nuanced understanding of the ASMR phenomenon as it is experientially understood and defined by the ‘ASMR Consumer’; thus, making for a valuable contribution to the broader quantitative literature, and I again thank the authors for their work and for revising the manuscript.

The manuscript is well presented and researched, and I feel it now nearly ready for publication. Though I list below a few very minor edits to the writing, I otherwise feel the manuscript ready for publication and am of the view that I will not need to review it further.

Abstract:

Start of line 31: change ‘experience and promote participant agency.’ to ‘experience of ASMR and to promote participant agency.’

Line 33: Change ‘trans-national’ to ‘transnational’

Line 34: Change ‘The transcriptions were analysed using…’ to ‘Interview transcripts were analysed using…’

Introduction:

Line 257: change [65-68) to [65-68]

Method:

Line 352: change ‘termination’ to ‘completion of the interview’

Final point: I appreciate the authors uploading copies of the transcripts to the Open Science Forum, however there is seven transcripts uploaded, instead of the supposed 6 (one for each participant). Indeed, it appears one of the transcripts ‘AL6’ has been uploaded twice.

7. PLOS authors have the option to publish the peer review history of their article (what does this mean?). If published, this will include your full peer review and any attached files.

Reviewer #1: **Yes: **Benjamin Lond

---

## [Editor Report · Acceptance letter]

10 Nov 2022

PONE-D-22-10021R1 

Beyond tingles: an exploratory qualitative study of the Autonomous Sensory Meridian Response (ASMR) 

Dear Dr. Hand:

I'm pleased to inform you that your manuscript has been deemed suitable for publication in PLOS ONE. Congratulations! Your manuscript is now with our production department. 

Kind regards, 

on behalf of

Dr. Jane Elizabeth Aspell 

Academic Editor

PLOS ONE